# The Reciprocal Longitudinal Relationship between Executive Dysfunction and Happiness in Korean Children

**DOI:** 10.3390/ijerph18157764

**Published:** 2021-07-22

**Authors:** Yoonhee Sung, Eunsil Choi

**Affiliations:** 1Department of Counseling Psychology, KC University, Seoul 07661, Korea; 0f30@kcu.ac.kr; 2Department of Family and Housing Studies, Yeungnam University, Gyeongsan-si 38541, Korea

**Keywords:** executive dysfunction, happiness, autoregressive crossed-lagged modeling, Korean children

## Abstract

The goal of this study was to examine the reciprocal longitudinal relationships between executive dysfunction and happiness for Korean children. We used data from the Panel Study of Korean Children (PSKC) conducted by the Korean Institute of Child Care and Education. A total of 1240 valid responses from the first to third grade in elementary school were analyzed using autoregressive crossed-lagged modeling. As a result, executive dysfunction and happiness were found to have reciprocal influences over the three time points. We also found that the cross-lagged effects of executive dysfunction and happiness were stronger than those of happiness on executive dysfunction. Clinical implications and limitations were discussed.

## 1. Introduction

The executive function (EF) has been considered a set of higher-order neuro-cognitive functions that are necessary to make decisions and to engage in purposeful and goal-directed behaviors [1,2,3]. Despite some arguments on the definition of EF, such as whether EF is a single or multi-dimensional component [4], it typically includes cognitive abilities to plan and organize situations, maintaining working memory and attention to specific activities, and the ability to control emotions and behaviors [5,6]. Thus, executive dysfunction in this study refers to cognitive difficulties in planning, organizing, attention, emotional and behavioral regulation.

The components of EF are closely and widely associated with cognition, emotion, and behaviors [7]. They are imperative for human development and adaptation, such as academic and social life [8]. Empirical research findings have supported the critical role of EF in a wide range of individuals’ functioning [3,9,10,11,12,13]. For example, Suchy showed that patients with brain lesions associated with executive functions presented more problems in behavioral and emotional regulation [3]. Gioia and his colleagues found that children with neurological deficits in EF had cognitive problems, such as difficulties in planning, organizing, and performing working memory [10]. In addition, Jahromi et al. demonstrated that preschoolers with higher executive dysfunction, assessed by four tasks, including PsyScope computer-based go [11] and day/night task [12], showed more negative emotional expression and used aggressive coping strategies more frequently [13]. The negative impact of executive dysfunction on one’s emotion and behaviors has been documented in several Korean studies as well. For instance, Park et al. found that parent ratings of children’s EF difficulties in second grade predicted problem behaviors in third grade [14]. Shin and her colleagues also showed that parent ratings of preschoolers’ executive dysfunction was positively associated with depressive symptoms and aggressive behaviors [15].

Many studies provide empirical evidence on the negative effects of executive dysfunction on socio-emotional and behavioral problems and psychopathology [9,10,16,17]. However, less is known about how executive dysfunction affects the positive aspects of socio-emotional functions. Given that positive and negative emotions are independent [18], research is needed to examine whether EF influences positive aspects of emotions, such as happiness.

Alternative theoretical perspectives suggest that emotions have a significant impact on cognition [19,20]. Emotions play a crucial role in motivating, organizing, guiding, and altering attention, perception, thought, and action, influencing how we perceive and respond to specific events [21,22,23]. It is well-documented that the regulatory aspects of emotions and cognitions, particularly emotion regulation and executive functioning, are closely interwoven [18,24,25,26]. Yet, Ferrier et al. suggested that the roles of emotion itself, especially positive emotions, have been overlooked in this area [27]. Therefore, examining the associations between positive emotions and EF would extend our understanding of how positive emotions are dynamically interrelated to cognition. Furthermore, an additional line of theoretical perspectives provides a rationale for the effects of positive emotion on cognition. According to Fredrickson’s broaden-and-build theory [28], positive emotions expand the scope of an individual’s cognition and attention, and can improve the efficiency and performance of EF [29].

Based on these theoretical perspectives, some researchers have investigated the effects of positive emotions on cognition. However, the results of previous studies have been inconsistent. For instance, Ferrier and his colleagues showed that the observational ratings of EF six months later also influenced emotionality in a sample of preschoolers, suggesting the importance of earlier positive emotions on complex cognitive processes [27]. In a sample of Korean children, Lee et al. found that children’s happiness significantly predicted EF [30]. Kwon et al. also showed that children’s general happiness was a significant predictor of EF, even when controlling for their self-esteem and internal and external problem behaviors [31]. In contrast, Finucane et al. found that there was no significant difference in EF among a happy, sad, and control condition, suggesting no effect of happiness on EF [32]. Other studies also supported the lack of significant effects of positive emotions on EF [33,34,35]. Furthermore, another experimental study found contradicting results that positive emotions in fact tended to impair EF [36]. Given these mixed findings, more research is needed to examine the role of happiness on executive function.

The aim of this study was to examine the bidirectional associations between executive dysfunction and happiness using three-year longitudinal data. We conducted autoregressive cross-lagged modeling (ACLM) to explore the bidirectional cross-lagged links between executive dysfunction and happiness over three years. ACLM is an advanced statistical strategy of comparing the directionality between two variables over time [37]. Examining executive dysfunction compared to happiness and happiness compared to executive dysfunction at the same time would allow us to compare the strengths of the pathways during childhood. Given that childhood is a critical period of EF development [38,39,40], examining the associations between executive dysfunction and happiness would not only affect adaptations during childhood, but also the development of EF and happiness in later lives.

## 2. Methods

### 2.1. Research Subjects

We used data from the Panel Study of Korean Children (PSKC) conducted by the Korean Institute of Child Care and Education. The PSKC is a national annual survey, running from 2008 to the present time on children’s development, family, and community environment. Data were collected by using stratified multi-stage sampling. The first stage sampled 30 medical institutions among those who reported more than 500 deliveries in 2006. The second stage involved sampling households who had a newborn baby in 2008 from the 30 medical institutions. The third stage sampled 2562 households who agreed to participate in the study. Out of this preliminary sample, 2078 households who responded to the first-year survey comprised the final panel of the PSKC. The data used in this present study came from the years 2015 (*N* = 1556, Time 1, first grade), 2016 (*N* = 1466, Time 2, second grade), and 2017 (*N* = 1394, Time 3, third grade), because 2015 was the first year that included the variable of executive dysfunction. After excluding missing data (*N* = 154), households who remained in 2017 and provided valid responses of children’s executive dysfunction and happiness across the three years were included in the final analytic sample (*N* = 1240); 619 were boys and 621 were girls.

### 2.2. Measures

#### 2.2.1. Executive Dysfunction

Executive dysfunction was assessed by using the Executive Function Difficulty Screening Questionnaire (EFDSQ) [41]. The EFDSQ was developed as a screening tool to identify children and adolescents with executive function difficulties in South Korea. The EFDSQ is a 40-item, three-point Likert scale ranging from 1 (“never”) to 3 (“very often”). The EFDSQ consists of four factors: planning/organization difficulties (11 items; an example is “He or she has difficulties doing things in order in a step-by-step manner”), behavior regulation difficulties (11 items; an example is “He or she has difficulties recognizing whether his or her behaviors would be annoying to other people”), emotion regulation difficulties (eight items; an example is “He or she has angry outbursts over little things”), and inattention (10 items; an example is “He or she is forgetful of his or her belongings and homework assignments”). The EFDSQ was completed by mothers on their children’s behaviors over the past six months. Cronbach’s alphas ranged from 0.84 to 0.91 for subscales and from 0.940 to 0.944 for the overall scale across time points.

#### 2.2.2. Happiness

Happiness was measured using six items that were developed by Youm et al. [42] and translated by the PSKC. Children were asked to provide their responses on how they feel about six parts of their life (schoolwork, the way they look, family, friends, school, and their life as a whole) on a scale of 1 (“not at all happy”) to 4 (“very happy”). An example item is, “How do you feel about your schoolwork?” A survey researcher visited participating households and asked the questions orally in person. Children indicated their responses by pointing out a picture that corresponds to their response on a face scale. The Cronbach’s alphas were 0.68 at Time 1, 0.72 at Time 2, and 0.75 at Time 3.

### 2.3. Data Analysis

Autoregressive cross-lagged modeling (ACLM) was conducted via Amos 23.0 to assess the reciprocal longitudinal relationships between happiness and executive dysfunction. The ACLM is useful for longitudinal analysis, in which the scores at time (t) depend on the scores at a prior time (t − 1) [43]. The ACLM provides information about autoregressive effects (the stability of the constructs from one time point to the next) and cross-lagged effects (the longitudinal prediction of one construct at one time point from the other construct at a prior time, while controlling for the autoregressive prediction of that construct).

To statistically evaluate the autoregressive and cross-lagged effects, we conducted model comparisons for three nested models to determine if the parameters were equal over time. First, a fully unconstrained model was used as the baseline model (Model 1), and then the constraints were imposed on subsequent models. Model 2 was constructed to examine the metric variance of measures by imposing constraints on the factor loadings to be equal across time. Model 3 was conducted to investigate the configural invariance by constraining each configural path loading to be equal across time.

The chi-squared value was limited due to its sensitivity to the sample size. Thus, we used the comparative fit index (CFI), the Tucker–Lewis index (TLI), and the root mean square error of approximation (RMSEA) to evaluate the goodness of fit. We used model evaluation criteria that indicated the potential for an acceptable fit (CFI, TLI > 0.90, RMSEA < 0.08) and excellent fit (χ^2^/df < 2.00, CFI, TLI > 0.95, RMSEA < 0.06 [44]. In determining evidence of invariance, two criteria were used. First, the model should show adequate fit indexes of CFI, TLI, and RMSEA. Second, differences in the CFI values between models should be less than 0.01 [45]. For happiness, three observed indicators (or parcels) were created to improve the model fit by reducing the number of parameters [46]. Items were assigned to parcels to equalize average loadings of each parcel on its corresponding latent variable [47]. For executive dysfunction, the average scores of the four factors were used as observed indicators.

## 3. Results

### 3.1. Preliminary Analyses

Data were examined for normality of distribution. Skewness and kurtosis ranged between ±1.45, so the data were assumed to be normally distributed [48]. Descriptive statistics and intercorrelations of the study variables are presented in Table 1. All correlations between study variables were significant and in the anticipated direction.

### 3.2. Testing Autoregressive Cross-Lagged Relations between Executive Dysfunction and Happiness

In the present study, executive dysfunction at Time 2 was predicted by both executive dysfunction at Time 1 and happiness at Time 1, and happiness at Time 2 was predicted by both happiness at Time 1 and executive dysfunction at Time 1. Similarly, executive dysfunction at Time 3 was predicted by both executive dysfunction at Time 2 and happiness at Time 2, and happiness at Time 3 was predicted by both happiness at Time 2 and executive dysfunction at Time 2. Before we tested the final model, we examined two assumptions of measurement invariance models. We first examined the metric invariance, which was required to ensure that items responded in the same way across time points. Next, we explored the assumption of configural invariance to ensure that our data supported identical patterns of fixed and nonfixed parameters across time points.

#### 3.2.1. Model Comparisons

Baseline model: Model 1 with freely estimated parameters showed excellent fit values. The chi-squared value was 338.026 (*df* 157), *p <* 0.0001; RMSEA = 0.031, CFI 0.986, TLI = 0.982. Model 1 indicated both the reciprocal relations between executive dysfunction and happiness over the three years, and the influences of each construct at one time point on the construct at the following year were significant.Model 2 (metric invariance model): To test the metric invariance of measures across the three time points, Model 1 (with no constraint) and Model 2 (with equality constraints on the factor loadings of executive dysfunction and happiness) were compared. In Model 2, the chi-squared value was 372.246 (*df* 167), *p <* 0.0001; RMSEA = 0.031, CFI = 0.984, TLI = 0.980. Model 2 also demonstrated an excellent fit to the data. When the two models were compared, the difference in the CFIs was negligible (ΔCFI < 0.01), indicating that the metric invariances of executive dysfunction and happiness were achieved. Given that Model 2 showed an excellent fit to the data and the CFI difference between the two models was less than 0.01, Model 2 with metric invariance was regarded as improved.Model 3 (configural invariance model): Invariance constraints were placed on every configural path of executive dysfunction and happiness (autoregressive and cross-lagged paths for executive dysfunction and happiness) in Model 3. Model 3 also yielded an excellent fit to the data. The chi-squared value was 386.683 (*df* 171), *p <* 0.0001; RMSEA = 0.032, CFI = 0.984, TLI = 0.980. The fit of Model 3 was excellent, and CFI was equal between Model 2 and Model 3, suggesting that Model 3 was improved compared to Model 2. Thus, we can conclude the configural invariance for executive dysfunction and happiness was established.Final Model: As a result of comparing the three nested models, Model 3 was found to be the optimal model. Model 3 was the most parsimonious, with equivalent fit indices. Thus, we selected Model 3 as the final model. The estimated values and factor loadings in Model 3 are presented in Figure 1. The results showed that both executive dysfunction and happiness had significant autoregressive effects across three years. All the autoregressive coefficients for executive dysfunction were positive and statistically significant, *β* = 0.792 and 0.792; all *p*-values < 0.001. The autoregressive coefficients for happiness were also all positive and statistically significant, *β* = 0.491 and 0.473; all *p*-values < 0.001.

#### 3.2.2. Reciprocal Relationship between Executive Dysfunction and Happiness

The cross-lagged analysis revealed that executive dysfunction and happiness had reciprocal influences across each time point. Specifically, the cross-lagged effects of executive dysfunction on happiness were significant, *β* = −0.149 and −0.147; all *p*-values < 0.001. The cross-lagged effects of happiness on executive dysfunction were also significant, *β* = −0.051 and −0.049; all *p*-values < 0.01.

#### 3.2.3. Comparing the Cross-Lagged Effects

A pairwise parameter comparisons test revealed that the cross-lagged effects of executive dysfunction on happiness were significantly stronger than the cross-lagged effects of happiness on executive dysfunction (critical ratio = −2.605, *p* < 0.01).

## 4. Discussion

The aim of the current study was to examine the bidirectional associations between executive dysfunction and children’s happiness using autoregressive cross-lagged modeling. We used the annual Panel Study of Korean Children for three years, which is representative data in Korea, and compared the strengths of executive dysfunction to happiness and happiness to executive dysfunction.

As anticipated, executive dysfunction at a previous time point was significantly associated with changes in children’s happiness, and children’s happiness at a previous time point was significantly related to changes in executive dysfunction. The results of this study provide evidence of the reciprocal associations between executive dysfunction and happiness in a sample of Korean children. Our findings are consistent with prior research on the effects of EF on socio-emotional functions [10,13,16,17] and behavioral problems [3,9,14,15]. In addition, the results are consistent with the literature on the effects of emotion on EF [20,26], particularly positive emotions [14,30,32]. The findings also provide additional empirical evidence to support the broaden-and-build theory of positive psychology. The broaden-and-build theory suggests that positive emotions can broaden the abilities of attention and thought–action repertoires [28]. By showing the effects of positive emotions on EF, this study extends the theory in a sample of Korean children.

Furthermore, we found that the cross-lagged effects of executive dysfunction on happiness were stronger than those of happiness on executive dysfunction. These findings suggest that executive dysfunction may lead to a negative cyclical relationship between executive function and happiness in childhood, such that children with executive dysfunction are more likely to be unhappy, which in turn would increase executive function problems. Compared to the rich literature on the effects of EF on problem behaviors and psychopathology [10,16,17], there are very few studies on the effects of EF on positive aspects of emotions and cognition in the development of children. One study found reciprocal associations between EF and emotionality in a sample of preschoolers [27]. Although they showed the parameter from EF at Time 1 to emotionality at Time 2 was larger than the parameter from emotionality at Time 1 to EF at Time 2, they did not examine if the strengths of the associations were statistically significant. The findings of the current study that showed the larger longitudinal effect of executive dysfunction highlight the crucial role of EF on the development of happiness during childhood.

Given the little research on the effects of EF on positive emotions and/or happiness, the findings suggest that more research is needed to understand the mechanisms of how executive dysfunction influences happiness during childhood. Considering that executive dysfunction is closely associated with emotion regulation [24,26] and social and emotional competences [17,49,50], those can be a plausible mechanism of the associations between executive dysfunction and happiness. Thus, further research that examines the mediating roles of emotion regulation and socio-emotional competences on the relationship between executive function and happiness would be beneficial.

The findings of this study have clinical implications. The findings suggest that the reciprocal process over time may provide a foundation for more adaptive development for children. For example, happy children are likely to have better executive function skills later, and children with better executive function tend to be happier. Conversely, the deficits in executive function or unhappiness may lead to negative outcomes in children’s development. Thus, paying close attention to children’s executive dysfunction and/or unhappiness is important to prevent a deviation-amplifying cycle during childhood. This is particularly true for children in South Korea. According to an international comparative study [42], the happiness index for Korean children (mean = 90.4) was lower than the average of OECD countries (mean = 100). Korean children’s unhappiness is closely associated with cultural context. South Korea is notorious for a competitive educational environment [51]. A lot of Korean children are exposed early to academic competition and suffer from academic stress, which was found to be a major source of unhappiness [52,53]. Considering our findings that suggest unhappiness during childhood can be a starting point of a deviation-amplifying process over time, interventions to improve children’s happiness is important. Given our findings on the stronger effect of EF on happiness, parents, teachers, and school counselors should help children learn and practice executive functioning skills, which in turn is likely to contribute to children’s happiness.

The findings of this study should be understood in consideration of the limitations. Children’s executive dysfunction was assessed by their mother’s report in our study. Given the emerging evidence that performance-based and rating scale measures of EF may assess different aspects of cognitive functioning [54], more research should be conducted using performance-based measures of EF, such as the Wisconsin Card Sorting Test [55] and the Stroop test [56]. Furthermore, we used a national sample, in which most of the participants do not have clinical levels of executive function problems. However, executive dysfunction and happiness may operate differently for clinical populations, such as children with brain impairment, attention deficit disorder, or autism spectrum disorder. Future research may examine the relationship between EF and happiness by comparing clinical and non-clinical samples. In addition, we did not consider gender in examining our research question. However, given prior research that indicates that gender plays a role in the way EF components relate to emotional and behavioral problems [57] as well as the level of happiness [58], examining gender differences in the reciprocal relationship between EF and happiness would be worth further research attention. Finally, the reciprocal relationship between executive dysfunction and happiness and stronger cross-lagged effects of executive dysfunction may be limited to Korean children. Thus, more research is needed to replicate the research questions of this study with different ethnic and age groups.

## 5. Conclusions

Our study with longitudinal data of Korean children found evidence that executive function and happiness are closely intertwined, indicating that lowering executive dysfunction is likely to enhance children’s happiness, and vice versa. Although both executive dysfunction and happiness have casual effects on each other, the path from executive dysfunction to happiness was found to be stronger than the path from happiness to executive dysfunction, suggesting that interventions on executive dysfunction may be relatively more effective. Our study has limitations in terms of measurement and generalizability, which should be addressed in future research.

## Figures and Tables

**Figure 1 ijerph-18-07764-f001:**
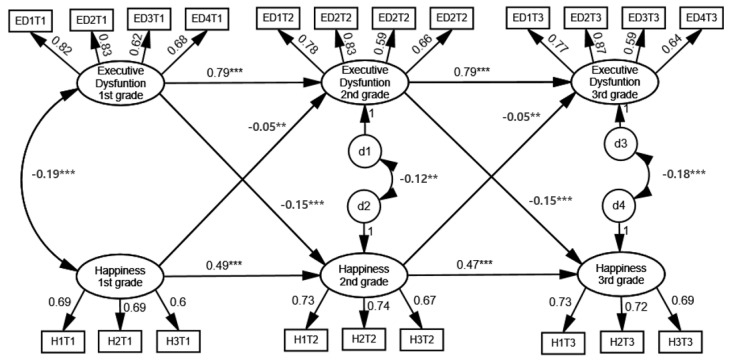
The fully constrained autoregressive cross-lagged model between executive dysfunction and happiness. ED = executive dysfunction, H = happiness, and T = time. ED1T1 = the first observed indicator for executive dysfunction at Time 1; H1T1 = the first observed indicator for happiness at Time 1. *** *p* < 0.001, ** *p* < 0.1.

**Table 1 ijerph-18-07764-t001:** The descriptive statistics and intercorrelations of the study variables (*N* = 1240).

	Mean	SD	1	2	3	4	5
1. Executive Dysfunction T1	1.44	0.31	-				
2. Executive Dysfunction T2	1.46	0.31	0.72 **	-			
3. Executive Dysfunction T3	1.47	0.38	0.70 **	0.75 **	-		
4. Happiness T1	3.26	0.48	−0.14 **	−0.16 **	−0.14 **	-	
5. Happiness T2	3.34	0.43	−0.17 **	−0.20 **	−0.19 **	0.35 **	-
6. Happiness T3	3.32	0.46	−0.21 **	−0.21 **	−0.27 **	0.26 **	0.44 **

** *p* < 0.01.

## Data Availability

The present study used publicly available datasets that is found in the website of Panel Study of Korean Children: https://panel.kicce.re.kr/panel/board/index.do?menu_idx=42&manage_idx=26, (accessed on 3 December 2020).

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
