# Peer review of "The Reciprocal Longitudinal Relationship between Executive Dysfunction and Happiness in Korean Children"

_ijerph, 2021, doi:10.3390/ijerph18157764_

Round 1

Reviewer 1 Report

This article presentes to us the results of study of Relationship  between Executive Dysfunction and Happiness in Korean Children.

The authors emphasize that "the aim of the current study was to examine the bidirectional associations between executive dysfunction and children’s happiness using autoregressive cross-lagged mod eling. We used the annual Panel Study of Korean Children for three years..." [P.5].

  • The article contains all the necessary sections:
    the material is presented logically and consistently;
  • methods of research and methods of data analysis are detailed;
    the study sample is characterized (all stages of the study are described);
  • the results of the study are summarized and conclusions are drawn.

On my own behalf, I want to note that such research has a practice-oriented focus.

It is clear that the results of this study can be published and will be of interest to other researchers for reading. 

Author Response

We really appreciate the comments from the reviewer. 

Thank you very much for highly evaluating our manuscript!

We tried to improve our manuscript following the comments from the reviewers.  

Reviewer 2 Report

  1. Paper's key findings and contribution

The paper details the bi-directional relationship between executive function (EF) and happiness in a sample of Korean children across three times points. Notably, it appears that EF is a stronger predictor of childhood happiness than happiness is a predictor of EF.  

The findings are well described and fit the summary provided by the authors.

  1. Strengths and Weaknesses

One of the main strengths of the paper is the sample size and modelling simplicity. The process of auto-regressive cross-lagged modelling (ACLR) is well described and justified for longitudinal data.

The paper is also well researched and by virtue of this the intentions of the research are clear.

A few minor alteration would help boost the paper’s impact. Firstly, please add details of each model’s composition in the Results section. Without this information it is unclear which variables (e.g., marital satisfaction) are having a significant impact on the variance during model comparisons.

Secondly, define the labels in Figure 1 in the figure caption; e.g., what do “ED1T1”, “d1” and “H1T1” represent? Providing this info will allow the reader to understand the model’s auto-regressive and cross-lagged effects.

Third and last, there are two points made in the discussion that need further embellishment. The first is mention of the broaden-and-build theory of positive psychology (line 221). This is mentioned at the end of a paragraph without much explanation. Please add details about the theory and how it is relevant in the context of your results.

The final point is that there is an opportunity to discuss childhood unhappiness levels in Korea from a cultural perspective here that is not exploited (line 251). By explaining the cultural differences underlying lower levels of childhood happiness in Korea you would give greater support for more research in this domain. It could be that the method you have deployed could be used to assess the significance of a new educational policy, and whether this has an effect on baseline and subsequent levels of EF and happiness.

Author Response

We appreciate the comments from reviewers and have strived to incorporate or respond to all of the suggestions. Please see the attachment.

Reviewer 3 Report

The present paper describes secondary analysis of longitudinal data from the Panel Study of Korean Children collected from 2015 to 2017 to examine associations between executive dysfunction and happiness.

General comments:

First, define executive dysfunction as used in this study more precisely. The introduction cited studies on a mix of different conditions representing varying components of EF. Please provide specific information regarding factors/tasks/variables included here. If summary scales or total scores were consistently used, please state this.

I am unfamiliar with the EFDSQ but assume it is somewhat similar to the CBCL. This is maternal self-report data. Happiness data appears to have been collected face-to-face by a researcher. Where was the study carried out (home, school, lab, other study site) and was it in person? Please elaborate.

What was controlled for? IQ? SES? Please define “constraints” in more detail 

How was missing data handled? This should be described in methods and results shown in results.

Please describe sample using numbers in the first paragraph of results. Were there 2,562 eligible households of which 1,240 were included in the present analyses? If there was no missing data, please state this. Please provide “n” for each and all analyses.

Throughout the manuscript, use the name of the 1st author et al., not the citation number, when referring to a person, team, or study (e.g., Smith et al. found that … [citation]) 

What does it mean that the sample is a “non-clinical sample”? Were families excluded if the child was diagnosed? Was there no ADHD or FASD? What kind of executive dysfunction was identified?  

Line 230 – comma

It would be interesting to see this data broken down by sex. Not necessary for this paper, but interesting.

Author Response

(The authors gave the same response as above.)
